# Quantification and Rehabilitation of Unilateral Spatial Neglect in Immersive Virtual Reality: A Validation Study in Healthy Subjects

**DOI:** 10.3390/s23073481

**Published:** 2023-03-27

**Authors:** Germain Faity, Yasmine Sidahmed, Isabelle Laffont, Jérôme Froger

**Affiliations:** 1EuroMov Digital Health in Motion, Université Montpellier, IMT Mines Ales, 34090 Montpellier, France; 2Physical Medicine and Rehabilitation, Nîmes University Hospital, Université Montpellier, 30240 Le Grau-du-Roi, France; 3Physical Medicine and Rehabilitation, Montpellier University Hospital, 34295 Montpellier, France

**Keywords:** stroke rehabilitation, unilateral spatial neglect, prism adaptation, subjective straight ahead, visual open-loop, pointing, serious game, virtual reality, HTC Vive, mocap

## Abstract

Unilateral spatial neglect is a common sensorimotor disorder following the occurrence of a stroke, for which prismatic adaptation is a promising rehabilitation method. However, the use of prisms for rehabilitation often requires the use of specific equipment that may not be available in clinics. To address this limitation, we developed a new software package that allows for the quantification and rehabilitation of unilateral spatial neglect using immersive virtual reality. In this study, we compared the effects of virtual and real prisms in healthy subjects and evaluated the performance of our virtual reality tool (HTC Vive) against a validated motion capture tool. Ten healthy subjects were randomly exposed to virtual and real prisms, and measurements were taken before and after exposure. Our findings indicate that virtual prisms are at least as effective as real prisms in inducing aftereffects (4.39° ± 2.91° with the virtual prisms compared to 4.30° ± 3.49° with the real prisms), but that these effects were not sustained beyond 2 h regardless of exposure modality. The virtual measurements obtained with our software showed excellent metrological qualities (ICC = 0.95, error = 0.52° ± 1.18°), demonstrating its validity and reliability for quantifying deviation during pointing movements. Overall, our results suggest that our virtual reality software (Virtualis, Montpellier, France) could provide an easy and reliable means of quantifying and rehabilitating spatial neglect. Further validation of these results is required in individuals with unilateral spatial neglect.

## 1. Introduction

Unilateral spatial neglect (USN) is a neuropsychological disorder characterized by difficulties in reporting, responding to, or directing attention to stimuli present on the side opposite to a brain lesion and not attributable to a sensory or motor deficit [1]. USN is prevalent in approximately 38% of individuals with right-hemisphere lesions and 18% of individuals with left-hemisphere lesions resulting from a stroke [2,3,4]. This behavioral disorder can have a profound impact on the functional prognosis of stroke patients, leading to poorer sensory–motor and cognitive performance and hindering their ability to perform daily living activities [5].

Several non-pharmaceutical rehabilitation techniques have been proposed to improve the functional prognosis of USN patients. However, the evidence supporting the efficacy of these rehabilitation protocols is limited due to the small number of studies that exist and methodological issues with these clinical trials [6]. One of the proposed techniques is prismatic adaptation, which involves performing a series of tasks while wearing prisms that shift the visual field by a specific amount. Immediate positive effects, including improvements in neglect (SBD 0.28, 95% CI −0.05 to 0.60) and activities of daily living (SBD 0.20, 95% CI −0.12 to 0.51), have been observed after removing the prisms following the session [6]. The original protocol for prismatic adaptation proposed two sessions, twice a day for two weeks, and symptom improvement was observed up to five weeks following exposure [7]. In healthy subjects, prismatic adaptation has been shown to induce a sensorimotor shift of approximately 4° to the left following exposure [8]. However, this technique requires specific equipment and a restrictive exposure protocol, limiting its use in rehabilitation centers [9]. Furthermore, measuring the effect of prismatic adaptation in patients is typically accomplished through pointing tasks, which require equipment that is rarely available in rehabilitation centers.

On the other hand, virtual reality has become a popular tool in rehabilitation centers for the treatment of motor and cognitive disorders, and has proven to be at least as effective as traditional rehabilitation methods in many areas [10], including upper-body rehabilitation [11,12,13,14,15,16]. For example, the HTC Vive is a popular virtual reality tool, particularly valued for its ease of use, tracking qualities, and low latency [17,18]. However, although it has millimeter-level validity, its reliability varies from millimeter-scale to meter-scale depending on experimental conditions [19,20,21,22], showing the need to evaluate this tool in a specific way depending on the targeted task. Finally, previous studies have shown that immersive virtual reality can produce results comparable to those obtained with real prisms in the treatment of unilateral spatial neglect [23,24,25]. However, to our knowledge, no study has directly compared the effects of real and virtual prisms on the mSSA test, which is the test that produces the largest effect size and has the strongest correlation with improvements in patients [26]. Therefore, the use of immersive virtual reality to assess and rehabilitate USN could overcome the limitations associated with traditional prismatic therapy. To address this need, we developed a software package for the quantification and rehabilitation of hemineglect using immersive virtual reality [27].

The primary objective of this study was to compare the effects of exposure to virtual and real prisms on two tests classically used to measure sensory–motor deviation: the subjective manual pointing straight ahead (mSSA) and the visual open-loop (VOL) tests [28]. We hypothesized that prism adaptation would induce negative deviation (leftward shift) in participants, independently of exposure and test modality (H1a). We also hypothesized that the induced deviation would persist over time and be visible in the retention test performed two hours after exposure (H1b). Finally, we hypothesized that the induced deviation under virtual conditions would be at least as large as the induced deviation under real conditions (H1c). Our second objective was to validate the HTC Vive^®^ tool for its use in measuring deviation angles against a validated motion capture system (CMS20s, ZEBRIS). We hypothesized that the deviation angle measured with the HTC Vive^®^ system would be the same as the deviation angle measured with the ZEBRIS system (H2).

## 2. Materials and Methods

### 2.1. Participants

Ten healthy participants (3 male and 7 female), aged between 23 and 39 years and including 2 left-handed individuals, were recruited for this study. Participants were excluded if they had any visual impairment that could not be corrected with glasses, spatial–temporal disorientation, a history of neurological disorders, uncontrolled psychiatric disorders, or orthopedic issues that might affect their test performance.

All participants provided both oral and written consent prior to their inclusion in the study. This study was conducted in accordance with the 1964 Declaration of Helsinki. The local ethics committee from Euromov Digital Health in Motion at Montpellier University approved the study (IRB-2106C).

### 2.2. Experimental Protocol

Each participant underwent three different test modalities to compare the effects of exposure to real and virtual prisms, as well as to validate the HTC Vive^®^ measurement tool against the ZEBRIS system. These modalities were:-The Real Set: measurements performed under real conditions with exposure to real prisms.-The Mixed Set: measurements performed under real conditions with exposure to virtual prisms.-The Virtual Set: measurements performed under virtual conditions with exposure to virtual prisms.

Each set included two blocks: an “exposure” block, followed by a “retention–recalibration” block performed two hours after the exposure block. Each block consisted of initial tests (mSSA, vSSA, VOL), 80 or 100 pointing movements with or without prisms depending on the block, and post-exposure tests (mSSA, vSSA, VOL) (Figure 1). To increase statistical power, each participant performed the three sets of tests, at one set per day, with a minimum interval of 48 h between two test days. The order of the tests was pseudo-randomized.

### 2.3. Description of Exposure and Tests Used

For the exposure to real prisms, participants were instructed to rest their chin on the fixed support and wear prismatic glasses which deflect the scene 10 degrees to the right. They were then asked to make a total of 80 rapid pointing movements using a controller towards a target located at 17 degrees of deviation to the right or left. The pointing movements were performed upon receiving oral instructions through a voice recording, with instructions for 40 pointing movements to the right and 40 to the left given in random order. During the pointing movements, the proximal course of the upper limb was concealed by a white support and only became visible to the participant at the end of the movement. The total exposure time for this task was approximately 2 min (Figure 2a).

In the case of exposure to virtual prisms, participants were instructed to rest their chin on the fixed support and wear a virtual reality headset which deviates the scene by 10 degrees to the right. They were also asked to perform a total of 80 rapid pointing movements using a controller towards a target located at 17 degrees of deviation to the right or left. The targets appeared randomly to the right or left once the previous movement was completed. During the pointing movements, the proximal path of the upper limb was computer-masked and only became visible to the participant in the virtual reality helmet during the last tenth of the movement. The total exposure time for this task was also approximately 2 min (Figure 2b).

For the manual subjective straight ahead (mSSA) task performed under real conditions, the same setup was used as for the exposure to real prisms. Participants were instructed to place their dominant hand against their sternum while holding the controller. Then, with closed eyes, they were asked to make a total of 10 pointing movements straight ahead. After each movement, they returned to the starting position in front of the sternum. The difference in angle between the median axis and the controller’s position at the time the controller touched the table was measured by the Zebris system in the plane of the table (Figure 3). The convention used for the measurements was that a right deviation was considered positive, while a left deviation was negative. The expected initial deviation was 0° on average (indicating no deviation), and the expected final deviation was 4° (representing approximately 40% of the prismatic exposure intensity).

In the case of the mSSA task performed under virtual conditions, the setup was the same as for virtual prism exposure. Participants were asked to place their dominant hand against their sternum while holding the controller. A black screen was projected in the headset, and they were instructed to make a total of 10 pointing movements straight ahead. After each movement, they returned to the starting position in front of the sternum. The measured variable was the same as that under real conditions, i.e., the difference in angle between the median axis and the controller’s position.

The visual subjective straight ahead (vSSA) test was conducted xclsuviely under virtual conditions. Participants were presented with a black screen with a green target that moved back and forth at a speed of 30 cm/s. They were instructed to press the button on the controller when the target was perceived to be in the center of the screen.

The visual open-loop pointing (VOL) test was conducted under both real and virtual conditions. For the real condition, a target was positioned on the median axis, and participants were instructed to visualize the target, and then close their eyes and point at it 10 times as accurately as possible. For the virtual condition, a scene containing a target positioned on the median axis was projected inside the headset, and participants visualized the target and pointed at it 10 times as accurately as possible. In both conditions, the difference in angle was measured between the axis of the target in the plane of the table and the position of the controller at the time when the controller touched the table.

The recalibration procedure was conducted to remove the aftereffect of the prisms and was performed only after the retention measurement, occurring 2 h after exposure. Participants randomly pointed at right or left targets 100 times without prismatic deviation.

### 2.4. Description of the Equipment

#### 2.4.1. Prism Exposure

For the real prism exposure, participants wore prismatic glasses that deviated their gaze by 10 degrees to the right, while a head support was used to mask the proximal stroke of their upper limb during pointing (Figure 2). The experimental setup was similar to that described in Rossetti’s article [29], where a horizontal board had two black stickers placed at 17 degrees from the sagittal plane which served as right and left targets.

The virtual prism exposure was performed using an HTC Vive^®^ immersive virtual reality system, which included a head-mounted display with a 110-degree field of vision and a 90 Hz refresh rate. The system was connected to a computer with sufficient power to support virtual reality games. Pointing was performed using an HTC Vive^®^ controller, while an HTC Vive^®^ tracker was used for calibration. The position of the different devices over time was captured using two infrared base stations included in the HTC Vive^®^ system.

The Protopointage software (Virtualis, Montpellier, France), developed in 2019 and later modified in 2022, was used to create a virtual environment for the study [27]. The participant was positioned in front of a table with a neutral background scene, and during the virtual prism exposure phase, the entire scene was shifted 10 degrees to the right (Figure 4). Targets randomly appeared on the table, both to the left and right, at 17 degrees from the participant’s median axis (corresponding to the targets on the physical support used in the study). During pointing, the participant’s virtual hand was concealed for 90% of the movement and only appeared in the final 10%, reducing real-time visual feedback.

#### 2.4.2. Measuring Tool

To compare the validity and reliability of the HTC Vive^®^ system, we recorded pointing movements using both a validated motion capture tool and the HTC Vive^®^ system. The CMS20s tool (Zebris Medical GmbH, Isny, Germany), which is widely validated and accepted as a gold standard in motion capture, allowed for a high spatial accuracy of better than 1.5 mm. Specifically, three markers were positioned on the controller to determine the position of its tip, and the sampling frequency was set to 50 Hz.

Prior to each participant’s session, the virtual plane of the table was calibrated according to the real plane of the table. The deviation angle provided by the Zebris mocap system, which served as the gold standard, was extracted from the motion data and compared with the deviation angle provided by the ProtoPointage software, which utilized the motion data from the HTC Vive^®^ system.

### 2.5. Data Processing

The data processing for this study was carried out using SciLab 6.1.0. For the Zebris motion capture system, the end of each pointing movement was defined as the moment when the position of the controller was closest to a virtual point located at the end of the table in front of the participant. The deviation angle was then computed as the difference between the final position of the controller and the median axis (Figure 3). The angles were directly exported from the ProtoPointage software (v0.2.1) for the HTC Vive^®^ system. The median angle of the 10 pointing movements was recorded for further analysis with both systems.

To account for the participants’ initial deviation when evaluating the aftereffect, the post-exposure deviation, which was used for hypothesis H1c, was corrected by subtracting the pre-exposure deviation from it (Equation (1)).
Final deviation (°) = post-exposure deviation (°) − pre-exposure deviation (°)(1)

### 2.6. Statistical Analyses

Statistical analyses were conducted using the R 4.2.1 software. To test for the presence of an aftereffect and retention effect for hypotheses H1a and H1b, a two-way ANOVA was performed with the within-factor “exposure”, having three modalities: real set, mixed set, and virtual set, and with the within-factor “time”, having four modalities: pre-exposure, post-exposure, 2 h retention, and post-recalibration. Tukey post hoc tests were then conducted to assess the differences between the modalities. For hypothesis H1c, the equivalence hypothesis was tested using a paired sample two one-sided test (TOST) procedure with an equivalence zone set at [−2, 2]°. One participant was excluded from the H1 analyses as they did not participate in the real set.

Furthermore, to evaluate the reliability of the HTC Vive^®^ system against Zebris for hypothesis H2, the intra-class correlation coefficient (ICC) and coefficient of determination (r^2^) were calculated. A Bland and Altman plot was also created to assess the validity of the Kinect by estimating a 95% confidence interval for the difference in means [30]. The ICC estimates were based on a one-way, consistent, single random effects model, where values below 0.5 indicated poor reliability, values between 0.5 and 0.75 indicated moderate reliability, values between 0.75 and 0.9 indicated good reliability, and values above 0.90 indicated excellent reliability [31]. The significance level for all tests was set at *p* < 0.05, and all reported coefficients of determination (r^2^) were statistically significant.

## 3. Results

### 3.1. Effect of Exposure Modality and Time

The results of the statistical analyses conducted on the effects of prism exposure on different tests of sensorimotor adaptation are presented in Figure 5. Specifically, the two-way ANOVA conducted on mSSA data showed no significant interaction effect between time and exposure modality for the angle of deviation (F(6) = 0.74, *p* > 0.05), and the exposure main effect was also not significant (F(2) = 1.07, *p* > 0.05). The time main effect, however, was significant, with only the pre- vs. post-exposure difference being statistically significant (F(3) = 5.83, *p* < 0.01; mSSA pre- vs. post-exposure difference = −4.22°, *p* < 0.001). The same pattern of results was obtained for the VOL test, with no significant interaction effect (F(6) = 0.42, *p* > 0.05) or exposure main effect (F(2) = 1.34, *p* > 0.05), and with only the pre- vs. post-exposure difference being significant (F(3) = 3.37, *p* < 0.05; pre- vs. post-exposure VOL difference = −2.81°, *p* < 0.05). For the vSSA test, the one-way ANOVA revealed no significant effect of time on the angle of deviation (F(3) = 0.87, *p* > 0.05).

Taken together, these results suggest that the prism exposure induced a 4.22° leftward deviation with the mSSA test and a 2.81° leftward deviation with the VOL test regardless of the exposure modality used. However, the exposure to prisms did not induce any significant difference in the vSSA test, nor did it result in any significant retention effect, regardless of the exposure modality or test used.

### 3.2. Equivalence Hypothesis

The present study investigated the equivalence of real and virtual prism exposure modalities on the deviation angle measured with the mSSA and VOL tests (Figure 6). For the mSSA test, a TOST equivalence test showed that the mixed modality was non-inferior to the real modality (t(8) = −2.17, *p* < 0.05), but the equivalence of treatments could not be concluded due to the failure to show the non-inferiority of the real modality compared to the mixed modality (t(8) = 0.37, *p* > 0.05), despite there being no significant difference between the 2 groups (t(8) = −0.89, *p* > 0.05). Using a second TOST equivalence test, the real modality was shown to be non-inferior to the virtual modality (t(8) = 2.22, *p* > 0.05), but the virtual modality was not shown to be non-inferior to the real modality (t(8) = −0.67, *p* > 0.05), leading to an impossibility of concluding on the equivalence of treatments despite there being no significant difference between the 2 groups (t(8) = 0.78, *p* > 0.05).

For the VOL test, a TOST equivalence test showed the non-inferiority of the mixed modality compared to the real modality (t(8) = −2.99, *p* < 0.001). However, again, the equivalence of treatments could not be concluded due to the failure to show the non-inferiority of the real modality compared to the mixed modality (t(8) = 0.59, *p* > 0.05), despite no significant difference existing between the 2 groups (t(8) = −1.20, *p* > 0.05). However, a second TOST equivalence test showed the non-inferiority of the virtual modality compared to the real modality (t(8) = −2.81, *p* > 0.05) and the non-inferiority of the real modality compared to the virtual modality (t(8) = 1.87, *p* < 0.05), allowing us to make conclusions on the equivalence of real prisms and virtual prisms in the deviation measured with VOL. Overall, the results suggest that virtual prisms may be a viable alternative to real prisms for inducing deviation aftereffects.

### 3.3. Is the Angle Given by the HTC Vive^®^ System the Same as the Angle Given by the Zebris?

The results of the present study reveal a high level of correlation between the angles of deviation which were obtained with the HTC Vive^®^ and Zebris systems, with an ICC of 0.95 and a coefficient of determination of 0.92. However, the intercept of the linear regression analysis (+0.53) suggests that the HTC Vive^®^ may have slightly underestimated the angle of deflection by approximately half a degree. Importantly, the slope of the regression analysis (+0.99) indicates that the error in the HTC Vive^®^ measurements did not depend on the size of the deflection. Bland and Altman plots further support these findings, indicating that on average, the deviation angles obtained with the HTC Vive^®^ were 0.52 ± 1.18° smaller than those obtained with the Zebris system. Additionally, 95% of the error in the deviation angles obtained with the HTC Vive^®^ fell within the interval of [−2.84; +1.79]° (Figure 7). These results suggest that while there may be a slight difference between the HTC Vive^®^ and Zebris systems in measuring deviation angles, the HTC Vive^®^ can still provide accurate enough measurements and be a viable alternative for clinical settings.

## 4. Discussion

### 4.1. For Hypothesis H1: “The Deviation Induced under Virtual Conditions Is at Least as Large as the Deviation Induced in the Real Condition”

In the first part of our study, we investigated the effect of virtual prisms on the deviation of the visual axis. Our findings revealed that exposure to virtual prisms resulted in a left deviation of 4.39° (±2.91°) for mSSA and 3.06° (±2.24°) for VOL. These values were compared with the deviation induced by real prisms, which was found to be 4.30° (±3.49°) for mSSA and 2.07° (±2.25°) for VOL. The results of our TOST equivalence tests indicated that virtual prisms were not inferior to real prisms, and there was no significant difference between the mixed set (real measurements) and the virtual set (virtual measurements). Our investigation also included the vSSA test, which did not show any significant pre-post difference which had been induced by the virtual prisms. Furthermore, we did not observe any retention effect 2 h after exposure, regardless of the exposure modality or test used. Overall, these findings suggest that virtual prisms could be a viable alternative to real prisms in inducing the deviation of the visual axis.

Our study results align with previous research, indicating that exposure to real prisms results in an aftereffect equivalent to 40% of the prismatic deviation in healthy [8] and stroke [32] subjects. Rode et al. have also shown that the aftereffect is weaker for the VOL test compared to the mSSA test, which we also observed in our study [28]. Several studies have attempted to reproduce the aftereffect in a virtual setting by using the HTC Vive^®^ system. Gammeri et al. gradually deviated subjects by up to 30° in a virtual environment and found an aftereffect of approximately 40–50% of the initial deviation during the VOL test, regardless of the exposure modality [23]. These results are promising, as the deviation measured during exposure in the virtual environment was larger than that obtained when exposure was performed with real prisms [28], which is likely due to the gradual deviation during exposure. Furthermore, the physics of real prisms does not allow for deviation >20° without distorting the image. Another study conducted using the HTC Vive^®^ system also found an aftereffect of approximately 5° for a 10° deviation on an mSSA test [24]. These studies suggest that an aftereffect can be observed after exposure to virtual prisms using the HTC Vive^®^ system. However, it is difficult to perform a rigorous comparison between virtual and real prism aftereffects since no control group which had been exposed to real prisms was included in these studies.

In the existing literature, only one study was identified that compared the effectiveness of virtual prisms to real prisms [25]. This study was conducted on healthy individuals and found a greater effect of virtual prisms compared to real prisms on the VOL test, with a virtual deviation of 4° versus 2° for real prisms. However, it should be noted that this study did not evaluate the mSSA test, which is the most strongly correlated with patient improvement and the test for which the effect of prisms is the greatest [26]. Furthermore, the magnitude of the virtual deviation used in this study was greater than the real deviation, and a post hoc transformation was performed to account for this difference. It is worth noting that the consequential effect is strongly correlated with the degree of prism adaptation [33], and thus caution is warranted in interpreting the results of this study. In contrast to this study and in agreement with the 2 studies cited previously [23,24], our study found an aftereffect of approximately 40% of the deviation intensity when measured with the mSSA test, regardless of the exposure modality (real and virtual). Additionally, our study found no significant difference between the mixed set and the virtual set, indicating that the effect of virtual prisms is transferable to real life, a necessary condition for demonstrating the usefulness of this technology.

Moreover, our study found no retention effect two hours after prism exposure in healthy subjects, in contrast to the findings of Rode et al. in patients with hemineglect [8]. This suggests that the duration of retention may be shorter in healthy subjects than in patients with USN [34]. It is possible that healthy individuals are able to recalibrate themselves more quickly during daily activities that require precision, such as using a smartphone or reaching for objects, thus reducing the retention effect. In contrast, hemineglect patients require prismatic exposure to recalibrate their visual attention and may be less likely to self-correct in real life, as the purpose of prism exposure is to recalibrate their perception and reduce errors in real-life tasks. Another possible explanation is that the neural pathways involved in prism adaptation differ between healthy individuals and stroke patients [34]. To investigate the existence and duration of deviation retention in healthy participants, future research could instruct participants not to engage in activities involving specific perceptual–motor coordination during the retention period to prevent unintended recalibration. Additionally, future studies should systematically assess retention at different time points to better understand the kinetics of the aftereffect, as this information is crucial for optimizing the use of prism adaptation in rehabilitation protocols.

### 4.2. Mechanism of Deviation

#### 4.2.1. The Specific Case of the vSSA

The results of our study indicate no significant aftereffect on the vSSA variable. One potential explanation for this finding is that the speed of the moving target was constant, which may have allowed participants to predict when the target would pass the middle of the screen. However, it is noteworthy that a study conducted by Rode et al. [28] on patients with USN also found no significant difference in vSSA. One possible explanation for this is that the perceptual–motor conflict required to elicit an error in healthy individuals may not be present in the vSSA task.

Indeed, in normal circumstances, spatial information from different sensory modalities is consistent. For instance, when individuals see a target in front of them and decide to point in its direction, they initiate the movement with their arm right in front of them, so as to reach the target with minimal error. However, this consistency between the various sensory modalities can be altered by optical devices. Prismatic glasses, for example, create a conflict between vision and proprioception when the hand is visible. During the initial exposure, due to the rightward-deviated vision induced by the glasses, the trajectory for pointing at the target is also shifted to the right (to correspond with the subject’s vision through the prismatic glasses). Because the visual and motor systems are no longer coordinated, the direct effects of the prisms result in an error of around 50 to 100% of the perturbation [29,35] (Figure 8).

If participants are able to perceive the deviation induced by prismatic glasses quickly enough, they can make corrections during pointing by using visual feedback [36]. This type of compensation does not lead to any specific adaptation of the individual to the prisms. However, if the movement is fast or if the initial trajectory of the movement is obscured, such as in our experiment, participants may not have sufficient time to correct their hand trajectory and will make errors due to the deviation (i.e., rightward error). With prolonged exposure (>30 pointing movements [37]), subjects can learn to compensate for the perceptual–motor conflict induced by the prismatic glasses through motor compensation. This involves changing the initial direction of the movement so that the hand moves slightly more to the left (i.e., in the direction initially intended) by integrating the deviation induced by the prisms [38]. This type of compensation is possible as the subject learns to move their hand while receiving erroneous visual information.

Once participants have adapted to the motor compensation required by the prisms, the removal of the glasses and the resumption of pointing movements will result in errors in the opposite direction (i.e., to the left). Having learned to compensate for the perceptual–motor conflict induced by the prisms by moving their hand further to the left than the target they see, participants will continue to do so, deviating the trajectory of their hand to the left with respect to the visual information they receive, believing it to be erroneous, even though the visual information is now correct. These consequential effects of prisms can account for approximately 40–80% of the perturbation in the opposite direction and depend on the magnitude of the prismatic deviation and the number of trials with the prisms, plateauing after 25–50 trials depending on the subject [33]. However, the perceptual–motor adaptation following prism wearing remains incomplete, even after numerous trials, and the reason for this remains unclear [32]. Motor learning experts explain this difference is being due to a separation of the explicit and implicit effects of prisms in learning, with only the implicit effects inducing aftereffects [39]. Recalibration of vision to its state prior to prism exposure can be achieved through several dozen pointing trials without the prismatic glasses.

During the vSSA test, participants are able to perceive their environment correctly without any deviation, as they are no longer wearing the prisms. However, the perceptual–motor conflict induced by the exposure aftereffects still disconnects participants’ motor actions from their vision. Since the moving target is visually localized and participants do not need to point at it as required in mSSA or VOL, there is no perceptual–motor conflict during the vSSA measurement in healthy subjects. As a result, there is no reason for a deviation in vSSA to be detected either before exposure, during exposure, or after exposure. This hypothesis may explain why there is an absence of an exposure aftereffect in the vSSA test in hemineglect patients, as observed in a previous study [28].

#### 4.2.2. The Importance of Visual Recalibration

In our 2021 pretests with 17 participants [40], we found that the perceptual–motor conflict was crucial in measuring the aftereffects of prism exposure. Initially, we had asked participants to perform the post-exposure mSSA test without reopening their eyes after the removal of the prismatic glasses in order to minimize visual cues and maintain the desired effects. However, under these conditions, no aftereffect was observed, which contradicted the previous literature (Figure 9). Further investigation revealed that participants needed to see the undeviated environment again after exposure to create the necessary perceptual–motor conflict. Without viewing the non-deviated scene, participants remembered the deviated environment and had become accustomed to the deviation, resulting in no perceptual–motor conflict during the mSSA test. By having participants view the non-deviated scene for a few seconds between exposure and the mSSA test, the expected perceptual–motor conflict was created, allowing for the measurement of the deviation.

In 2022, during our use of the updated Virtualis software, we encountered a similar issue as in our 2021 pretests. Specifically, we found that without the visualization of the undeviated environment between exposure and the mSSA measurement, no aftereffect was observed during the virtual reality protocol. To address this issue, we contacted the developer of the software and requested that they add a feature to allow the participant to view an undeviated environment for 3 s after virtual exposure. This additional viewing time enabled the participant to create the necessary perceptual–motor conflict, similar to what occurs when the glasses are removed under classical conditions.

### 4.3. For Hypothesis H2: “The Angle of Deflection Measured by the HTC Vive^®^ System Is the Same as the Angle Measured by the ZEBRIS System”

The second part of our study demonstrates the high quality of metrology achieved by the HTC Vive^®^ virtual reality system when compared to the results of conventional Zebris CMS20s motion capture system, having an interclass correlation coefficient (ICC) of 0.95 and a measured difference in angle of 0.52° (±1.18°, 95% CI = [−2.84, 1.79]). To our knowledge, no previous research has ever compared virtual reality measurement tools with conventional measurement tools. These findings indicate that the virtual reality system exhibits excellent metrological qualities in comparison to the Zebris CMS20s system.

### 4.4. Perspectives

In addition to being at least as effective as the conventional prism exposure method, the immersive virtual reality rehabilitation system offers numerous advantages. Firstly, the proximal path of the upper limb, which is crucial in perceptual–motor recalibration, can be easily masked by computer adjustment, unlike the classical method which employs a head rest. Secondly, the virtual reality system allows for multiple computer adjustments, facilitating personalized rehabilitation programs that can adapt to the patients’ disorders and progress. Finally, since the sensory–motor parameter measurement and exposure can be carried out under virtual conditions, the measurement procedure is significantly simplified. This facilitates follow-up, making it possible to use the system daily during patient rehabilitation.

A future study is planned to validate the effectiveness of the virtual reality rehabilitation system on patients with USN. If the study shows positive results on sensory–motor parameters, a personalized rehabilitation protocol could be developed by adjusting the degree of prismatic deviation and the frequency of exposure to maintain the effect over time [41]. It is also important to measure the improvement in activities of daily living to determine the long-term effectiveness of the virtual prisms. The effectiveness of rehabilitation methods for patients with hemineglect has been a subject of debate due to the low methodological quality of existing studies [6]. However, recent research by Chen et al. (2022) [41] challenges this conclusion and suggests that further investigation is warranted.

## 5. Conclusions

In conclusion, our study shows that immersive virtual reality rehabilitation is at least as effective as conventional prism exposure in inducing sensory–motor deviation in healthy subjects. Moreover, the virtual reality system has many advantages over the conventional method, including the ability to easily adjust the degree of prismatic deviation, personalize the rehabilitation program, and simplify the measurement of sensory–motor parameters. Our study also demonstrates that the HTC Vive^®^ virtual reality system has excellent metrological qualities compared to conventional measurement tools. Our findings suggest that virtual reality rehabilitation has great potential as a tool for treating patients with USN, and that further research is needed to validate its effectiveness in the target population. The success of this rehabilitation tool could provide a promising new avenue for treating this disabling condition, leading to improvements in the quality of life for patients and their caregivers.

## Figures and Tables

**Figure 1 sensors-23-03481-f001:**
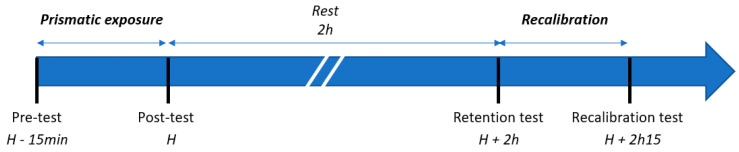
Protocol time course for prismatic exposure and recalibration. This figure illustrates the timeline of tests conducted in the study, which involves two main steps: prismatic exposure and recalibration, with a 2 h rest period in between. Each step comprises pre-test, exposure, and post-test steps. For the first step, the pre-test and post-test are labeled accordingly. For the second step, the pre-test is referred to as the “retention test”, and the post-test is known as the “recalibration test”.

**Figure 2 sensors-23-03481-f002:**
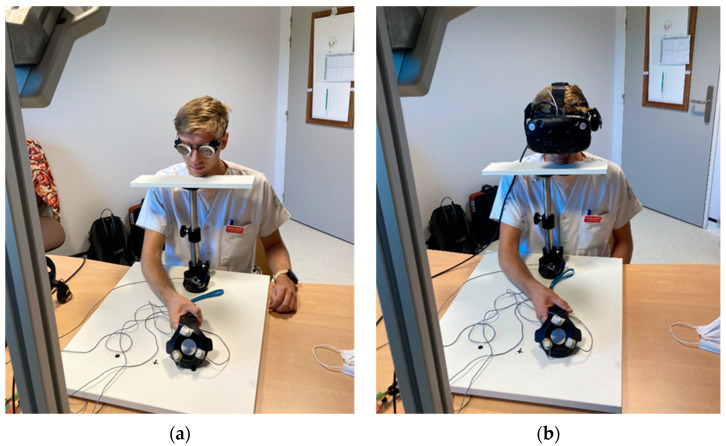
(**a**) Participant during exposure to real prisms; (**b**) Participant during exposure to virtual prisms.

**Figure 3 sensors-23-03481-f003:**
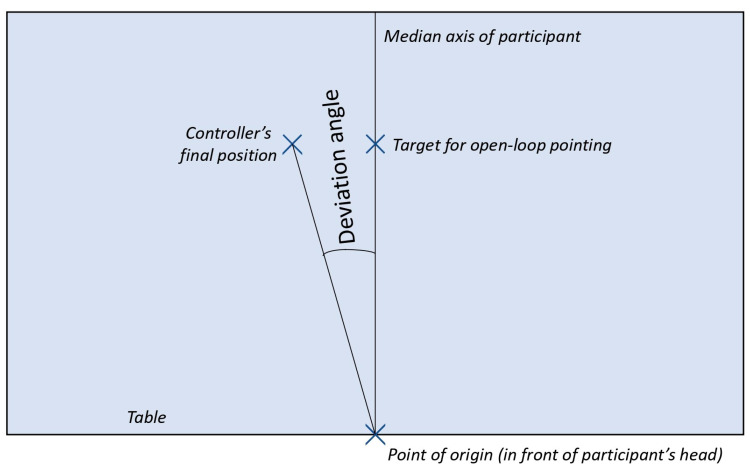
Method of calculating the angle of deviation.

**Figure 4 sensors-23-03481-f004:**
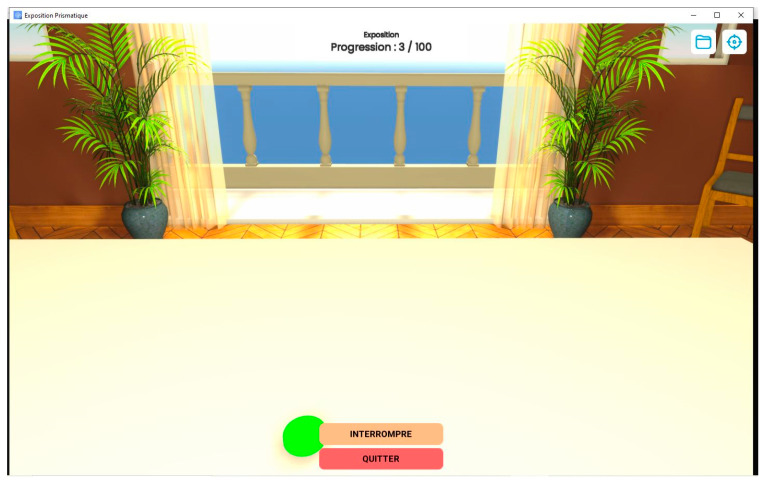
Interface of the Virtualis game. The participant visualizes the scene with a 10° deviation to the right. A green target is shown here, located at 17° to the left. When the participant points at the target, a new target will appear randomly on the left or right.

**Figure 5 sensors-23-03481-f005:**
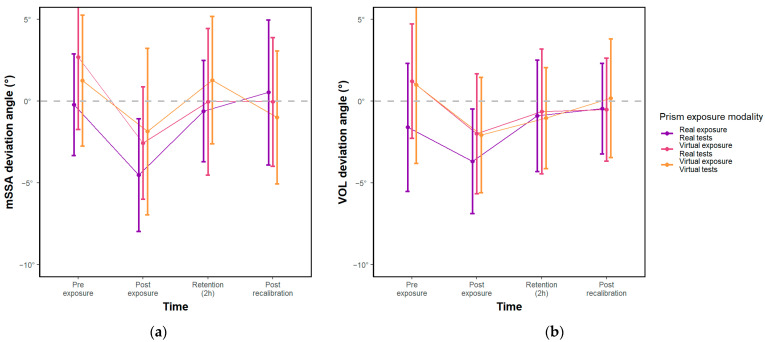
(**a**) Deviation evaluated in mSSA according to the 3 modalities and the 4 times tested; (**b**) Deviations evaluated in VOL. Dots correspond to means, vertical bars to standard deviations.

**Figure 6 sensors-23-03481-f006:**
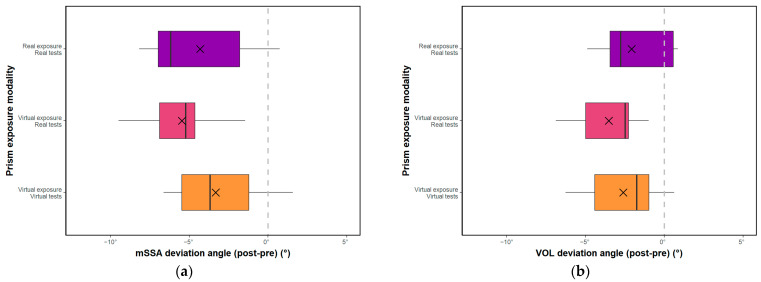
(**a**) Post-effect-evaluated in mSSA according to the 3 test-exposure modalities; (**b**) Post-effect-evaluated in VOL according to the 3 test-exposure modalities. Crosses correspond to means, and vertical bars to medians. Boxes span 25th to 75th percentiles. Whiskers cover the 95% confidence interval.

**Figure 7 sensors-23-03481-f007:**
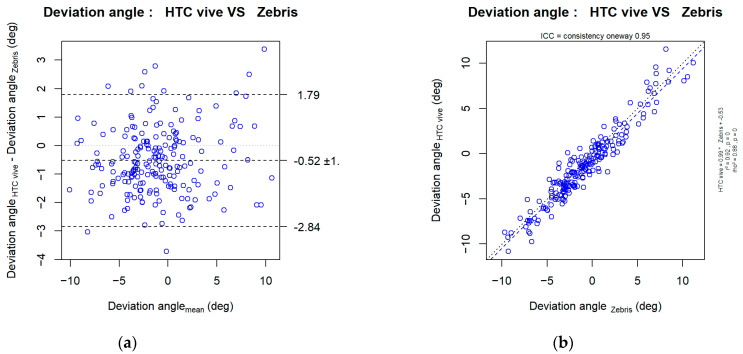
(**a**) Bland–Altman plot; (**b**) Regression plot (right panel) of the angle measured by the HTC Vive^®^ system versus the Zebris motion capture system considered as the gold standard. The angles of deviation obtained with the HTC Vive^®^ and Zebris are highly correlated, but a slight underestimation was made with the HTC Vive^®^.

**Figure 8 sensors-23-03481-f008:**
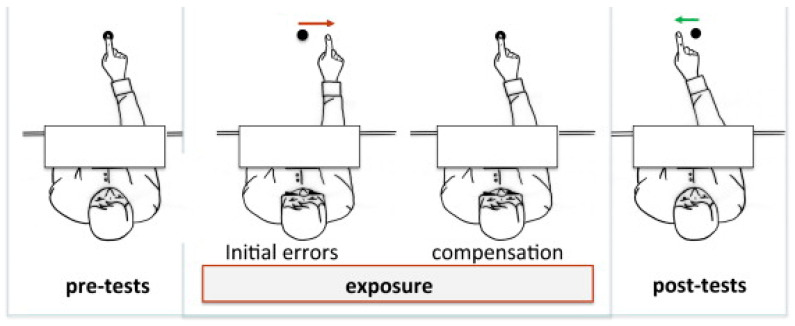
Pre-test (**left**), exposure (**middle**) and aftereffect (**right**). Adapted from Rode et al. [28]. During the pre-test phase, no deviation is observed (**left** panel). During the exposure phase, the subject wears prismatic glasses that deviate the vision by 10 degrees to the right. When the subject points at the target, the motion is shifted to the side of the optical deviation and towards the virtual target (**middle left** panel). The pointing deviation is gradually corrected and the deviation is completely compensated after a sufficient number of trials (**middle right**). After removal of the prismatic glasses, the compensation remains while the prismatic deviation is no longer present, producing a shift of motion in the opposite direction to the optical deviation, called after-effects (**right** panel).

**Figure 9 sensors-23-03481-f009:**
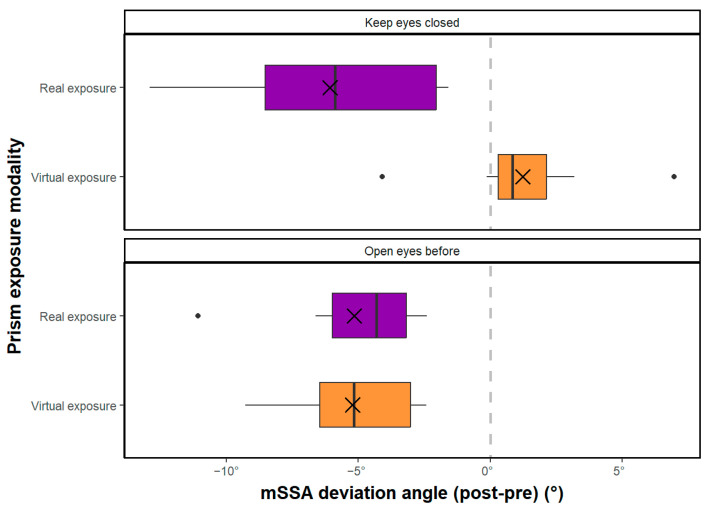
Results of our pre-tests. The deviation effect was only present under real conditions when participants opened their eyes to see the scene after exposure.

## Data Availability

The data presented in this study are openly available in the open science framework depository at https://osf.io/zgpuk.

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
