# Peer review of "Quantification and Rehabilitation of Unilateral Spatial Neglect in Immersive Virtual Reality: A Validation Study in Healthy Subjects"

_sensors, 2023, doi:10.3390/s23073481_

Round 1

Reviewer 1 Report

     In this manuscript, the so called Quantification and rehabilitation of unilateral spatial neglect by prismatic adaptation in immersive virtual reality: a validation study of the 

HTC vive in healthy subjects is analyzed and discussed.

Nevertheless, the commented key issues must be re emphasized to the authors, due to the low content quality of this eidtion.

  (1)  In the abstract, no any numerical gain for the proposed unilateral spatial neglect by prismatic adaptation in immersive virtual reality is included. The authors must provide the several key performance improvements to identify the superiority of the developing  unilateral spatial neglect by prismatic adaptation in immersive virtual reality compared to the well discussed conventional Models.

 (2)  Almost 17 of the all 28 references are published more than four even ten years ago which fail to represent the edge cutting development of this research direction.

  (3) The table of all parameters for  the  proposed unilateral spatial neglect by prismatic adaptation in immersive virtual reality  must be included in the modification of this work and the reference and

  setting support of all parameters and the datasheets must be labelled to prove the physical implementability and repeatability.

  (4) The too apparent similarity is derived between the abstract and the conclusion. The several distinct numerical results must be presented 

  in the conclusion, compared to the abstract. Moreover, the conclusion is too long and not compact enpugh.

  (5)  The numberical comparison must be made between the proposed unilateral spatial neglect by prismatic adaptation in immersive virtual reality  with the conentional schemes in several performance metrics

  (6) This is no any mathematical desciption of the so called proposed unilateral spatial neglect by prismatic adaptation in immersive virtual reality included in this manuscript. 

     The complete mathematical architecture must be included to numerially evaluate the system performance.

  (7) There is no any stroke patient is introduced to investigate the system performance and adaptability.

Before ore all above modifications are made, I cannot recommend this manuscript to publish. 

Reviewer 2 Report

In this paper the authors have compared the effects of exposure to virtual and real prisms on two tests classically used for measuring sensory-motor deviation. The proposition of the study is interesting; but there are some important drawbacks which must be addressed and Clarity of presentation can be improved by giving more detail in the paper before publication in Sensors Journal. Therefore, this reviewer considers that the manuscript does have some important drawbacks that are listed below.

1.     The title should be shortened; currently it is very lengthy and looks informal.

2.     It is necessary to rephrase the abstract because it has not been written well. For example, line no 24 to 26 are not appropriative statements. 

3.     I suggest to authors to divide an introduction section in subsections i.e., Unilateral spatial neglect (USN), analytical reasoning that why virtual technologies are used in the rehabilitation of stroke patients. In this manner, the reader will easily understand the objective of this study.

4.     The should mention their hypothesis in bullet forms and also describe in conclusion that what they achieved by their hypothesis.

5.     The authors should mention the structure/organizing of the paper at the end of introduction.

6.     More detail require for Figure 1.

7.     The most important thing is the lack of latest related works (systems); the authors have not included the literature review regarding the related system.

8.     The authors have not clearly defined the actual problem statement based on the related work (literature review) that why the authors study or their proposed system is necessary. Therefore, the authors should mention clearly the problem statement at the end of related work.

9.     The methodology of the part of the analysis is not clear, nor is the analysis of the validity and their limitations.

10.  Most importantly that how the participants are invited for evaluation? Detail is necessary.

11.  The conclusion is described vaguely and did not reflect the overall objective of the study.

12.  The English writing should be improved.

Round 2

Reviewer 2 Report

Accepted
